# Causal Inference of Central Nervous System-Regulated Hormones in COVID-19: A Bidirectional Two-Sample Mendelian Randomization Study

**DOI:** 10.3390/jcm12041681

**Published:** 2023-02-20

**Authors:** Yuxuan Sun, Ziyi Ding, Yawei Guo, Jinqiu Yuan, Chengming Zhu, Yihang Pan, Rui Sun

**Affiliations:** 1Scientific Research Center, Sun Yat-sen University, The Seventh Affiliated Hospital, Shenzhen 518107, China; 2The Herbert Wertheim School of Public Health and Longevity, University of California San Diego, La Jolla, CA 92093, USA; 3School of Public Health, Sun Yat-sen University, Guangzhou 510275, China

**Keywords:** Mendelian randomization, CNS-regulated hormones, COVID-19

## Abstract

We assessed the causal association of three COVID-19 phenotypes with insulin-like growth factor 1, estrogen, testosterone, dehydroepiandrosterone (DHEA), thyroid-stimulating hormone, thyrotropin-releasing hormone, luteinizing hormone (LH), and follicle-stimulating hormone. We used bidirectional two-sample univariate and multivariable Mendelian randomization (MR) analyses to evaluate the direction, specificity, and causality of the association between CNS-regulated hormones and COVID-19 phenotypes. Genetic instruments for CNS-regulated hormones were selected from the largest publicly available genome-wide association studies of the European population. Summary-level data on COVID-19 severity, hospitalization, and susceptibility were obtained from the COVID-19 host genetic initiative. DHEA was associated with increased risks of very severe respiratory syndrome (odds ratio [OR] = 4.21, 95% confidence interval [CI]: 1.41–12.59), consistent with multivariate MR results (OR = 3.72, 95% CI: 1.20–11.51), and hospitalization (OR = 2.31, 95% CI: 1.13–4.72) in univariate MR. LH was associated with very severe respiratory syndrome (OR = 0.83; 95% CI: 0.71–0.96) in univariate MR. Estrogen was negatively associated with very severe respiratory syndrome (OR = 0.09, 95% CI: 0.02–0.51), hospitalization (OR = 0.25, 95% CI: 0.08–0.78), and susceptibility (OR = 0.50, 95% CI: 0.28–0.89) in multivariate MR. We found strong evidence for the causal relationship of DHEA, LH, and estrogen with COVID-19 phenotypes.

## 1. Introduction

Coronavirus disease 2019 (COVID-19), caused by severe acute respiratory syndrome coronavirus 2 (SARS-CoV-2), has infected more than 452 million individuals and resulted in more than 6 million deaths worldwide as of 12 March 2022 [1]. SARS-CoV-2 infection has high variability in susceptibility, hospitalization, and severity, with clinical severity ranging from asymptomatic or mild upper respiratory illness to moderate and severe diseases such as respiratory failure, acute respiratory distress syndrome, and multi-organ failure with fatal outcomes [2]. A report on COVID-19 suggests that 80% of infections are mild or asymptomatic, while 20% are moderate to severe [3]. At the early infection stage, the innate immune system recognizes SARS-CoV-2 and induces proinflammatory cytokines to remove the viruses [4]. However, SARS-CoV-2 disrupts the normal immune response in severely and critically ill patients, leading to an uncontrolled inflammatory response and cytokine storm, resulting in systemic and local damage [4,5]. Understanding the host factors influencing this infectious disease is crucial for elucidating the viral life cycle, identifying risk factors, drug development, and disease prevention.

CNS-regulated hormones have received special attention in the context of COVID-19, considering their modulatory effects on cytokines. The gender disparity, with reports that men experience more severe metabolic syndromes and have higher mortality than women do, suggests that metabolome influenced by sex hormones such as estrogen and testosterone may play a role in COVID-19 morbidity and mortality [6,7,8]. Androgens may be associated with an increased risk of COVID-19 due to their modulatory effects on angiotensin-converting enzyme 2 (ACE2) and transmembrane serine protease 2 (TMPRSS2), the two key factors associated with the entrance of SARS-CoV-2 [9]. Indeed, anti-androgen therapy in patients with prostate cancer has reportedly protected them against infection with SARS-CoV-2 [10]. A recent study also suggested a causal effect of increased testosterone levels with a higher risk of COVID-19 hospitalization and severe disease [11]. In contrast, a retrospective study found that older females who received estradiol therapy had a lower fatality risk than older females who did not receive the therapy, indicating a potential immune protective effect of estrogen in females [12]. Meanwhile, in a study comparing critically and non-critically ill patients, low insulin growth-like factor-1 (IGF-1) levels showed a potential association with severe forms of COVID-19 [13].

Despite the possible effect of CNS-regulated hormones on COVID-19, there appears to be a reverse directional effect of COVID-19 on hormones. A cohort study found that as the severity of COVID-19 increased, the concentration of dehydroepiandrosterone (DHEA) significantly increased [14]. However, another observational study of hospitalized patients did not find any difference in DHEA levels between males and females with severe COVID-19 [15]. Another study on thyroid function found that thyroid-stimulating hormone (TSH) levels in COVID-19 patients were obviously lower than those in non-COVID-19 patients, which suggested the possible effects of COVID-19 on TSH [16]. 

Although observational studies provide some evidence for the conferral of hormonal effects on COVID-19, the liability of these findings is challenged by confounding factors, reverse causation, and other factors [17]. Therefore, the causal impact of hormones on COVID-19 risk has not been ascertained. Mendelian randomization (MR) uses genetic variants following Mendel’s law of independent assortment as unconfounded proxies of an adjustable exposure to examine whether the exposure has causal effects on an outcome, overcoming the limitations of observational studies [18].

In this study, we used two-sample MR univariate and multivariable analyses to examine potential causal associations between three types of COVID-19 (very severe respiratory syndrome, hospitalization, and susceptibility) and eight CNS-regulated hormones [IGF-1, estrogen, testosterone, DHEA, TSH, thyrotropin-releasing hormone (TRH), luteinizing hormone (LH), and follicle-stimulating hormone (FSH)]. We also performed a bidirectional analysis to detect the possible reverse causal relationship between CNS-regulated hormones and COVID-19. Anti-Coronavirus vaccines such as the BioNTech/Pfizer, Moderna, and Oxford/AstraZeneca vaccines have been proven to have prophylactic and therapeutic effects, which were highly recommended for protection [19,20,21]. However, more therapeutics for COVID-19 are still needed. By assessing and establishing the causal role of CNS-regulated hormones, possible hormonal therapies could be applied as therapeutics for COVID-19 to reduce mortality and morbidity in COVID-19-infected patients.

## 2. Material and Methods

### 2.1. CNS-Regulated Hormones and COVID-19

Single-nucleotide polymorphisms (SNPs) associated with the levels of the eight selected hormones were selected based on summary-level data from either the Integrative Epidemiology Unit (IEU) open genome-wide association study (GWAS) project [22,23] or the largest meta GWAS of thyroid-related traits [24]. Strict European samples were selected as exposure sets, part of which were from the UK Biobank. The sample sizes of the exposure set varied from 468,343 to 1000. Genetic instruments for TSH were extracted from a meta-analysis of thyroid hormones in 26,420 participants of European ancestry [25]. Estradiol is the most potent form of estrogen in the human body; thus, it was used to represent the function of estrogens [26]. In addition, DHEA-sulfate (DHEA-S) was adopted to represent DHEA because it circulates at a far higher concentration than DHEA in the blood [27]. All GWAS summary data were adjusted for age, sex, and other study-specific covariates. Detailed information is provided in Appendix A. 

Summary statistics for the three COVID-19 phenotypes were obtained from the release of the five COVID-19 Host Genetics Initiative (HGI) GWAS meta-analysis [28]. The COVID-19 HGI is a consortium of scientists from over 54 countries working collaboratively to investigate human genetic variation related to very severe respiratory syndrome (cases: 5582; controls: 709,010), hospitalization (cases: 17,992; controls: 1,810,493), and susceptibility (cases: 87,870; controls: 2,210,804) (Appendix A). All individuals included in our analysis were of European ancestry. Very severe respiratory syndromes were cases that ended with respiratory support or death from COVID-19. Hospitalizations were cases that involved hospitalized patients. Susceptibility was defined as cases that tested positive, including laboratory-confirmed COVID-19 using reverse transcription-quantitative polymerase chain reaction (RT-qPCR) or serological testing, clinically diagnosed COVID-19, and self-reported COVID-19. In the GWAS meta-analysis, single-variant association analyses of COVID-19 traits were performed after adjusting for age, age^2^, sex, age × sex, top principal components for ancestry, and study-specific covariates of each contributing cohort. All studies contributing data to the analyses were approved by the relevant ethics committees.

### 2.2. Selection of Genetic Instruments for CNS-Regulated Hormones and COVID-19 Traits

Genetic instruments were selected from the exposure data with genome-wide significance (*p* < 5 × 10^−8^). SNPs were excluded if (1) they were not present in the outcome GWAS summary data, (2) SNPs in linkage disequilibrium were within a 1-Mb genomic distance (R-squares above 0.01) using the 1000 Genome European reference panel and retained the SNP with the lowest *p*-value, and (3) minor allele frequency was greater than 0.42 for non-inferable palindromic alleles. For exposures with a relatively small number of SNPs, such as DHEA-S and TRH, if an SNP was absent in the outcome dataset, an LD proxy SNP was searched using the 1000 Genome European panel to replace the missing SNP [29]. Exposure and outcome GWAS summary statistics were harmonized by aligning the effect alleles. The number of genetic instruments used in bidirectional MR analyses is shown in Appendix A. The proportion of variance was calculated using the sum of the variance explained by individual SNPs divided by the variance of phenotype [30] (Appendix A).

### 2.3. Bidirectional Two-Sample MR and Statistical Analysis

We conducted bidirectional two-sample MR analyses. CNS-regulated hormones were considered exposures in the forward direction, and COVID-19 severity, hospitalization, and susceptibility were the outcomes. Conversely, COVID-19 severity, hospitalization, and susceptibility were exposures, and CNS-regulated hormones were outcomes in the reverse direction. The procedures for genetic instrument selection were similar in both directions of analysis.

We applied three MR methods to estimate causal effects: inversed-variance weighted (IVW) regression/Wald ratio, weighted median (WM), and MR-Egger regression. IVW regression was the primary approach used to derive causal estimates, where at least two exposure SNPs were available for analysis. Under the assumption of no horizontal pleiotropy, the IVW approach could generate an unbiased causal estimate [31]. The Wald ratio method was used when only one instrumental SNP was available for analysis. When the pleiotropy assumption was not satisfied, the WM method would have a lower bias but a higher type I error rate than the IVW method and would therefore be used to further verify the inference effects [31]. The MR-Egger test was used to assess horizontal pleiotropy. The intercept of MR-Egger represents the average horizontal pleiotropic effect across the exposure genetic instruments, where the slope of the regression was the casual estimate [32]. 

Multivariable Mendelian randomization (MVMR) analyses were conducted to account for potential confounding factors due to the possible genetic correlation of sex hormones, including estradiol, testosterone, DHEA-S, LH, and FSH [33]. Multivariable MR analysis could account for pleiotropy in a univariable MR analysis by including a set of covariates and could distinguish between the direct effects of the exposures on the outcome and the total effects inclusive of mediators, which cannot be accessed in the univariable MR analysis [34]. Multivariable lasso was used to perform MVMR analyses, as multivariable lasso is mostly a robust multivariate method, accounting for outliers and pleiotropy caused by invalid instruments [34]. After combining SNPs of sex hormones, we clumped SNPs for linkage disequilibrium r^2^ < 0.01 within a 1-Mb genomic distance, excluded overlapping SNPs, and removed SNPs lacking linkage information. The remaining SNPs were used for MVMR analysis. We also conducted MVMR in the reverse direction to account for possible confounding due to the genetic correlation of three types of COVID-19 on CNS-regulated hormones, showing evidence of univariable MR associations with at least one COVID-19 trait. For traits with insufficient SNPs (*N* ≤ 3) to perform MVMR analyses, genetic instruments were selected with a suggestive genome-wide significant *p*-value threshold (*p* < 1 × 10^−5^). 

Sensitivity analyses were performed to verify the robustness of the results. Heterogeneity among the SNPs included in each analysis was examined using Cochran’s Q test. We used the MR-Egger regression intercept test and MR pleiotropy residual sum and outlier (MR-PRESSO) global test to check for horizontal pleiotropy. The MR-PRESSO method was used to check and correct for outliers [35]. *I*^2^*_GX_* statistic was used to examine the measurement error in the SNP-exposure association [36]. The strength of the selected SNPs was assessed by F-statistic [37] calculated using the total sample size, the number of SNPs used, and the proportion of variance explained. 

The significance threshold was corrected for multiple testing with each MR analysis using the Bonferroni method (*p*-value < 0.05/number of exposures). In the sensitivity analysis, as the *p*-value evaluated the precision of the effect estimation and given the large number of statistical tests performed, we evaluated each result with a *p*-value < 0.05 individually by considering heterogeneity, pleiotropy, and removal of outliers. All data analyses were performed in R [38] using the TwoSampleMR [23], MendelianRandomization [39], and MR-PRESSO [35] packages.

## 3. Results

### 3.1. Forward MR Results: Impact of CNS-Regulated Hormones on COVID-19 

#### 3.1.1. Univariable MR Analysis Using CNS-Regulated Hormones as Exposures

As shown in Figure 1, genetically predicted DHEA-S concentrations were nominally associated with very severe respiratory syndrome (OR = 4.21, 95% CI: 1.41–12.59, *p* = 0.010). This means that each standard deviation increase in DHEA-S level would dramatically increase the odds of having very severe respiratory syndrome by 4.21-fold. Additionally, DHEA-S was significantly associated with hospitalization (OR = 2.31, 95% CI: 1.13–4.72, *p* = 0.021). Along with the insignificant but positive association between DHEA-S and susceptibility (OR = 0.94, 95% CI: 0.67–1.31, *p* = 0.703), the results showed a possible trend between DHEA-S and COVID-19; DHEA-S had an increased OR and significance level for the three COVID-19 phenotypes (susceptibility, hospitalization, and very severe respiratory syndrome). There was evidence of a negative association between LH level and very severe respiratory syndrome (OR = 0.83, 95% CI: 0.71–0.96, *p* = 0.013). In the susceptibility analysis, we did not find significant causal effects of CNS-regulated hormones on COVID-19 phenotypes (Appendix A). The F statistics of all instrument variables were > 10, with the proportion of variance explained ranging from 0.069% to 17.879%, indicating the absence of weakness in the selected instruments (Appendix A). 

#### 3.1.2. Multivariable MR Analysis with Sex Hormones

After adjusting for other sex hormones, genetically predicted DHEA-S was still significantly associated with an increased risk of very severe respiratory syndrome at a nominal significance level (OR = 3.72, 95% CI: 1.20–11.51, *p* = 0.023) (Table 1). Although the association between DHEA-S and hospitalization and susceptibility became insignificant, the trend matched what we found in the univariable analysis; DHEA-S had an increasing effect size and significance level as COVID-19 severity increased. In addition, the effect estimates of LH on very severe respiratory syndrome were invalidated by conditioning the genetic effects of other sex hormones. However, we observed significant associations between estradiol and very severe respiratory syndrome (OR = 0.09, 95% CI: 0.02–0.51, *p* = 0.006), hospitalization (OR = 0.25, 95% CI: 0.08–0.78, *p* = 0.016), and susceptibility (OR = 0.50, 95% CI: 0.28–0.89, *p* = 0.019). The different results of LH and estradiol levels between univariate and multivariable analyses indicated that sex hormones interacted with each other during COVID-19. Notably, estradiol consistently increased the effect size and significance level as COVID-19 severity decreased, as in the univariate analysis, which further supported progressive interactions with COVID-19. Other than estradiol and DHEA-S, we did not find any significant associations between CNS-regulated hormones and COVID-19. Meanwhile, in the MVMR analysis, hormones other than testosterone also exhibited a similar increasing OR trend as DHEA-S in the univariate analysis, suggesting possible progressive immune interactive mechanisms during COVID-19. 

### 3.2. Reverse MR Results: Testing for Bidirectionality Using COVID-19 Traits as Exposures

We did not find evidence supporting reverse causality between DHEA and all three COVID-19 phenotypes. However, there was evidence of the promoting effects of very severe respiratory syndrome on IGF-1 (OR = 1.01, 95% CI: 1.00–1.02, *p* = 0.040), and susceptibility on testosterone (OR = 1.05, 95% CI: 1.03–1.06, *p* < 0.001) after outlier correction (Appendix A).

In MVMR analysis, after adjusting for other COVID-19, very severe respiratory syndrome still had promoting effects on testosterone (OR = 1.11, 95% CI: 1.03–1.19, *p* = 0.003) and hospitalization was found to have significantly decreasing effects on testosterone (OR = 0.89, 95% CI: 0.84–0.94, *p* < 0.001) (Appendix A). 

### 3.3. Sensitivity Analysis

#### 3.3.1. Test for Horizontal Pleiotropy

In univariate analysis with the COVID-19 phenotype as the outcome, the MR-Egger regression intercept test did not find any horizontal pleiotropy in the association between eight CNS-regulated hormones and three COVID-19 phenotypes (all *p*-values > 0.05) (Appendix A). However, the MR-PRESSO global test suggested horizontal pleiotropy in the association of IGF-1 and testosterone with COVID-19 phenotypes (*p*-value < 0.05).

In the bidirectional analysis using CNS-regulated hormones as outcomes, there was no evidence of horizontal pleiotropy in any associations between COVID-19 and hormone traits using the MR-Egger regression intercept test (all *p*-values > 0.05). The MR-PRESSO global test suggested potential horizontal pleiotropy effects associated with very severe respiratory syndrome COVID-19 and susceptibility to IGF-1 and the three COVID-19 phenotypes with testosterone (all *p*-values < 0.05). Apart from IGF-1 and testosterone, no horizontal pleiotropy effect was found in either direction for any of the three COVID-19 phenotypes, which indicates the validity of the significant findings in the univariable MR analysis (Appendix A). 

#### 3.3.2. Test for Heterogeneity of Instruments

As shown in Appendix A, in the univariate analysis with COVID-19 severity as an outcome, heterogeneity was found in IGF-1 and testosterone estimates. Cochran’s Q test indicated heterogeneity in the association of IGF-1 with very severe respiratory syndrome (*p* = 0.001) and susceptibility (*p* = 0.007). Heterogeneity was only associated with very severe respiratory syndrome (*p* < 0.001).

For the analysis using COVID-19 as exposure, heterogeneity was found in the association among very severe respiratory syndrome, susceptibility, and IGF-1 and in the association between very severe respiratory syndrome and testosterone (*p* < 0.001 for all analyses) (Appendix A). 

#### 3.3.3. Test for Measurement Error

In the analysis using COVID-19 severity as an outcome, only TSH was found to have a low I^2^_GX_ statistic associated with COVID-19, which indicated a potential bias to MR-Egger estimators (Appendix A).

In the analysis using very severe respiratory syndrome as exposure, only DHEA-S had an I^2^_GX_ statistic > 0.900, which suggested that MR-Egger estimates of all other hormones could be biased by measurement errors (Appendix A). We did not find any measurement errors in the hospitalization analysis based on the I^2^_GX_ statistics. Like very severe respiratory syndrome, almost all the associations between susceptibility and hormones were biased by measurement errors, except for DHEA-S.

## 4. Discussion

### 4.1. Main Findings

Bidirectional two-sample MR analyses using large publicly available genomic datasets were used to analyze the causal association between the genetic liability of CNS-regulated hormones and COVID-19 susceptibility, hospitalization, and severity. We found strong evidence supporting the causal associations of DHEA and LH with very severe respiratory syndrome COVID-19 and the association of DHEA with hospitalization due to COVID-19. We did not find evidence supporting an association between CNS-regulated hormones and COVID-19 susceptibility. In MVMR analyses, we found evidence of a negative association between estradiol and COVID-19 susceptibility, hospitalization, and severity, adjusted for other sex-related hormones. In the reverse direction, we found that very severe respiratory syndrome was associated with IGF-1 levels, and COVID-19 susceptibility was associated with testosterone levels. Together with previous observational studies, we revealed possible causal associations between CNS-regulated hormones, especially sex hormones, and COVID-19 severity, suggesting the possibility of applying hormonal therapies to COVID-19 severe patients.

The effects of DHEA on COVID-19 are contradictory. On the one hand, DHEA can exert immunomodulatory and anti-inflammatory functions, which means that DHEA could have potential protective effects. Conversely, as a type of androgen, DHEA is suspected to be associated with an increased risk of COVID-19. Our analysis clearly indicated that DHEA is associated with an increased risk of COVID-19, especially the risk of very severe respiratory syndrome COVID-19. As mentioned before, DHEA, as a type of androgen, has regulatory effects on the ACE-2 receptor and TMPRSS2, promoting the fusion of the SARS-CoV-2 virus into host cells, thereby increasing the susceptibility and severity of COVID-19 [40,41]. Indeed, with its androgenic effect, DHEA also acts as a powerful inhibitor of glucose-6-phosphate dehydrogenase (G6PD) [42]. G6PD deficiency has been demonstrated to enhance cell infection with human coronavirus 229E (HCoV 229E) [43]. Considering that SARS-CoV-2 is also a human coronavirus, DHEA may greatly increase the risk of COVID-19 due to its inhibition of G6PD. 

In contrast to androgens, estrogen is the main reason why the female immune system responds more efficiently to pathogens [44]. In the analysis, when adjusting for all other sex hormones, estrogen was found to have significantly decreasing effects on all types of COVID-19, which supports the possible protective effects of estrogen and indicates the potential for the implementation of hormonal therapy [45]. In fact, clinical trials are in progress to test the effect of additional estrogen on COVID-19 patients [46].

Moreover, the effects of DHEA and estrogen all displayed a progressively increasing effect size and significance level as the COVID-19 severity increased. Two possible mechanisms may play a role in this pattern. The first is the difference in viral load between severe and mild cases. A study of the viral load during COVID-19 reported that compared with mild cases, patients with severe COVID-19 tend to have higher viral loads [47]. Considering the effects of DHEA in enhancing the fusion of viruses into host cells, DHEA might result in a higher risk for people with higher viral loads than those with lower viral loads. Another possible explanation is the mechanism of COVID-19. The key factor affecting severity is cytokine storms. Estrogen has powerful anti-inflammatory effects due to its inhibition of proinflammatory cytokines such as IL-1β, IL-6, and others [48]. Meanwhile, estrogen has strong regulation effects on several aspects of metabolism, especially glucose metabolism [49]. Glucose metabolism dysregulation has been found to be associated with the severity of COVID-19 as viral replication depends on glucose [50]. Furthermore, the cytokine storm is also associated with the dysregulation in glucose metabolism, which contributes to the severity of COVID-19 [50]. A study also showed that severe cases have higher blood glucose levels compared with mild cases of COVID-19 [51]. As COVID-19 severity increases, uncontrolled immune responses produce more proinflammatory cytokines as well as increased glucose levels; therefore, the regulatory effect of estrogen becomes more powerful, which leads to increasing effect size and significance level. However, due to the regulatory effect of estrogen, hormonal therapy should be used carefully in patients with dysmetabolic conditions such as diabetes.

We did not find any association between testosterone and the three types of COVID-19. A recent MR study on serum testosterone and bioavailable testosterone found that only the bioavailable testosterone level was associated with a higher risk of hospitalization due to COVID-19, indicating that different forms of testosterone may have different effects on COVID-19 [11]. Indeed, we found the promoting effect of COVID-19 susceptibility on testosterone levels. Some studies have observed decreased testosterone levels in severe cases. However, other studies have also proposed that the inflammatory response caused by SARS-CoV-2 viruses may trigger testosterone secretion to initiate an anti-viral reaction considering the anti-inflammatory and immune-modulatory effects of testosterone [52,53,54]. In addition to testosterone levels, we also observed that COVID-19 severity was associated with increased levels of IGF-1. However, a phenome-wide association study [55] found that genetically determined severe COVID-19 was associated with lower IGF-1 levels. The contradiction of the results suggests that further studies and an understanding of IGF-1’s mechanisms and pathways are needed.

### 4.2. Additional Findings

Our findings also elucidate the possible interactive effects of sex hormones on COVID-19. We found significant effects of LH on very severe respiratory syndrome COVID-19. A recent COVID-19 case–control study involving 89 men with COVID-19 suggested that impaired testosterone in infected males may stimulate LH release to maintain testosterone levels based on negative-feedback regulation [53]. Analysis using COVID-19 as exposure indicated that COVID-19 was associated with increased testosterone levels. Combined with the anti-inflammatory function of testosterone, increased LH levels could decrease the risk of severe COVID-19. Another possible reason could be the interaction between LH and estradiol. Univariate analysis did not find a possible causal association between estradiol and COVID-19. 

In contrast, after adjusting for other hormones in multivariable analysis, estradiol was found to be significantly negatively associated with all three types of COVID-19 with alterations in the effect direction and significance level of LH. LH is the upstream hormone of testosterone and estrogen, which controls the secretion and transformation of these hormones based on the negative-feedback loop. The true effect of LH may be influenced by both estrogen and testosterone levels. 

### 4.3. Strengths and Limitations

The strengths of our study include the integration of univariate, multivariate, and bidirectional MR analyses using genetic instrument variables drawn from the largest GWASs to date. We also included a variety of CNS-regulated hormone levels and COVID-19 phenotypes. Furthermore, follow-up sensitivity analyses such as the MR-Egger intercept test, the Cochran’s Q test demonstrated the robustness of our findings. The F-statistic of the instrument variables also illustrated the strength of our selected SNPs. Finally, applying multivariable and univariable approaches is an advantage highlighted by the converging findings for COVID-19 phenotypes. We minimized the chances of population stratification between exposure datasets and outcome datasets by using GWAS of the European population for CNS-regulated hormones and by excluding the 23andMe cohort from COVID-19 datasets.

Our study has the following limitations. First, it is possible that the severity of COVID-19 and the immune response of patients were infected by different strains of COVID-19. Our study aimed to assess the general causal relationship between CNS-regulated hormones and COVID-19 instead of assessing the association with certain strains of COVID-19. Thus, we did not include the strain of COVID-19 in the study. Secondly, the selected genetic instruments generally explain a small to moderate proportion of the variance in exposure, although well within the typical ranges for complex traits [56]. Investigations of genetic variants associated with CNS-regulated hormones are needed. In addition, some exposures had only one or two SNPs, which were not suitable for sensitivity analyses. The use of weak genetic instruments may limit our ability to detect subtle, causal associations. However, the roughly similar pattern between univariate and multivariate analysis, and the high F-statistic, suggested the robustness and reliability of our results. Considering the possible sex effects, we used the UK Biobank sex-stratified data to assess the effects of testosterone, estrogen, IGF-1, and TSH on COVID-19. The results showed an effect direction consistent with that of the univariate two-sample MR analysis (Appendix A). 

## 5. Conclusions

In conclusion, our study found a possible causal association between CNS-regulated hormones and COVID-19. In contrast to previous observational studies focusing on a single hormone, such as testosterone or estrogen, our study was the first to use MR analysis to investigate the causal association between CNS-regulated hormones and COVID-19 severity, hospitalization, and susceptibility. We found that genetically predicted DHEA and LH had significant causal associations with COVID-19; and that significant effects of estrogen on COVID-19 were observed after adjusting for other hormones, supporting the feasibility of hormonal therapy for COVID-19. Future studies may need to be designed to further understand the mechanisms and pathways underlying the causal relationship between CNS-regulated hormones and COVID-19, with the goal of developing potential therapeutic strategies. 

## Figures and Tables

**Figure 1 jcm-12-01681-f001:**
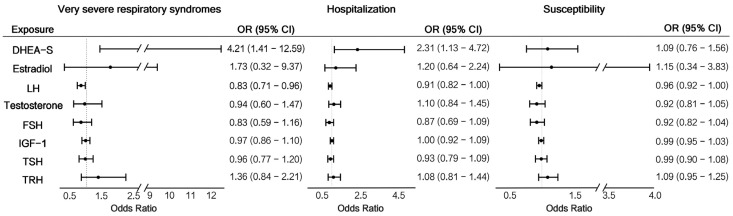
MR analyses the association of CNS-regulated Hormones and COVID-19 phenotypes.

**Table 1 jcm-12-01681-t001:** MVMR analysis of CNS-regulated hormone SNPs and COVID-19 severity and susceptibility.

	Very Severe Respiratory Syndrome		Hospitalization		Susceptibility	
Exposure	OR (95% CI)	*p*-Value	OR (95% CI)	*p*-Value	OR (95% CI)	*p*-Value
DHEA-S	3.72 (1.20–11.51)	0.023	1.69 (0.83–3.42)	0.144	0.94 (0.67–1.31)	0.703
Estradiol	0.09 (0.02–0.51)	0.006	0.25 (0.08–0.78)	0.016	0.50 (0.28–0.89)	0.019
LH	1.23 (0.81–1.86)	0.343	1.16 (0.88–1.52)	0.296	1.08 (0.94–1.24)	0.251
FSH	1.11 (0.84–1.48)	0.464	1.03 (0.85–1.24)	0.751	1.00 (0.91–1.10)	0.947
Testosterone	0.42 (0.17–1.03)	0.058	0.74 (0.41–1.33)	0.312	0.76 (0.56–1.02)	0.067

## Data Availability

GWAS summary data of seven hormone traits are available in the IEU open GWAS project (Accessed 2 December 2021; https://gwas.mrcieu.ac.uk/) with GWAS ID as ukb-d-30770_irnt for IGF-1, ukb-d-30800_irnt for Estradiol, ukb-d-30850_irnt for Testosterone, met-a-478 for DHEA-S, prot-a-3102 for TRH, prot-a-529 for LH, and prot-c-3032_11_2 for FSH. GWAS summary data of TSH are collected in a GWAS meta-analysis of thyroid-related traits (20). Summary statistics (release 5) of GWAS meta-analysis for the three COVID-19 phenotypes are available in the COVID-19 Host Genetics Initiative (HGI) at https://www.COVID19hg.org/results/r5/ (Accessed 2 December 2021).

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
