# Peer review of "Causal Inference of Central Nervous System-Regulated Hormones in COVID-19: A Bidirectional Two-Sample Mendelian Randomization Study"

_jcm, 2023, doi:10.3390/jcm12041681_

Round 1

Reviewer 1 Report

I really enjoyed reading your work. It is an interesting paper and a lot of information could be used by researchers to complete their work. I have only a few commentaries:

1. Why did you focused on the European population as genetic instrument? Isn't the Asian population better fitted for this study? My comment does not refer to a limitation of the study, I am just curious as to why.

2. How many days of hospitalization were considered?

3. In the discussion part, line 232-234, you are missing the verb of the sentence (probably "was used").

4. Line 242: I am not sure about the truthfulness of the hormonal therapy as "possible treatments" of Covid-19. Please re-phrase it.

5. Line 291: So, in the end, is there an association between Covid-19 and estrogen? Which statistical analysis is trustworthy: the univariate analysis or the mendelian one? Discuss a bit about the advantages of the mendelian analysis in direct coroboration with your study.

6. In the Author Contribution statement, there are only three authors listed with a direct contribution to the article. What is the role of the other 4 authors? (if they did nothing significant, maybe you should put them in the Acknowledgement and not as co-authors; in contrast, if they actively participated in the research, they should have their role shown in the statement).

Reviewer 2 Report

Dear authors,

I haver reviewed the manuscript entitled “Causal Inference of Central Nervous 2 System-regulated Hormones in COVID-19: A Bidirectional 3 Two-sample Mendelian Randomization Study ” by Yuxuan Sun and coworkers.

In this paper the authors have used two-sample MR univariate and multivariable analyses to examine potential causal associations between three types of COVID-19 and eight CNS-regulated hormones [IGF-1, estrogen, testosterone, DHEA, TSH, TRH, LH, and FSH. In addition, they have done a bidirectional analysis to detect the possible reverse causal relationship between these CNS-regulated hormones and COVID-19. They have found very interesting associations between some of these hormones (DHEA, LH, and estrogen) and the different severity of COVID-19 infected patients.

However, I believe the authors should address in the manuscript the following:

-There is no mention of whether the severity of the COVID-19 infection is related with the patients having been immunized by vaccines (or even the number of shots or origin of the vaccine) which were available by march 2022, the date until the authors state they have recollected the data.

I consider this is a variable that must be taken into account. At least the authors should mention it is a limitation of the study if they do not consider it.

-There is no mention of the stain of the COVID-19 variable the patients had. Different variables of the SARS-CoV-2 virus have been shown to more or less aggressive.  This fact could also be associated with the response of the patient to the infection. I believe the authors should comment in the discussion.

Reviewer 3 Report

please find the comments attached

Reviewer 4 Report

In their study entitled „Causal Inference of Central Nervous 2 System-regulated Hormones in COVID-19: A Bidirectional 3 Two-sample Mendelian Randomization Study“ Sun et al used Mendelian randomization analysis to study severity, hospitalization and susceptibility to COVID-19 as variables on one side in relation to IGF-1, estradiol, testosterone, DHEA, TSH, 68 thyrotropin-releasing hormone (TRH), luteinizing hormone (LH), and follicle-stimulating hormone (FSH) on the other. The concentrations of hromones were predicted based on SNPs in relevant alleles available from patient and control DNA sequencing data. Three MR methods to estimate causal effects were applied inversed-variance weighted (IVW) 116 regression/Wald ratio, weighted median (WM), and MR-Egger regression. Their results show that DHEA and LH have a causal relationship with severe COVID-19 and DHEA with COVID-19 related hospitalization, while estradiol had a negative association with susceptibility, hospitalization, and severity. In the reverse analysis, very severe respiratory syndrome was associated with IGF-1 levels, and 239 COVID-19 susceptibility was associated with testosterone levels.

The study is interesting and contributes to better understanding about, not only which hormones are related with specific outcomes, but what might be the causal relationship between hormone concentrations and outcomes. The study is solidly written, data is presented clearly. 

The study is based on SNPs in patient sequencing data for prediction of hormone levels, which allows large patient and control groups for analysis. However, DNA sequence might not completely accurately reflect hormone levels in patients and controls, as other factors might influence this as well. The authors should comment on this.

The authors also perform analysis in the reverse direction, indicating that severe respiratory syndrome was associated with IGF-1 levels, and 239 COVID-19 susceptibility was associated with testosterone levels. However, SNPs in the genomes of patients were present before they encountered SARS-CoV2. Does it make biological sense to do the reverse analysis?

Remove the word „title“ from the title.

Round 2

Reviewer 3 Report

The manuscript has been improved. In the current form deserves publication. Best wishes.